# NEGOTIATING TEAM FORMATION USING DEEP REINFORCEMENT LEARNING

## ABSTRACT

When autonomous agents interact in the same environment, they must often cooperate to achieve their goals. One way for agents to cooperate effectively is to form a team, make a binding agreement on a joint plan, and execute it. However, when agents are self-interested, the gains from team formation must be allocated appropriately to incentivize agreement. Various approaches for multi-agent negotiation have been proposed, but typically only work for particular negotiation protocols. More general methods usually require human input or domain-specific data, and so do not scale. To address this, we propose a framework for training agents to negotiate and form teams using deep reinforcement learning. Importantly, our method makes no assumptions about the specific negotiation protocol, and is instead completely experience driven. We evaluate our approach on both non-spatial and spatially extended team-formation negotiation environments, demonstrating that our agents beat hand-crafted bots and reach negotiation outcomes consistent with fair solutions predicted by cooperative game theory. Additionally, we investigate how the physical location of agents influences negotiation outcomes.

## 1 INTRODUCTION

Multiple agents inhabiting the same environment affect each other, and may gain by coordinating their actions. Indeed, many tasks are effectively intractable for any single agent, and so can only be solved by a team of collaborators. Examples include search and rescue (Kitano et al., 1999), multi-robot patrolling (Agmon et al., 2008), security (Tambe, 2011) and multiplayer first-person video games (Jaderberg et al., 2018). Despite the need to cooperate, stakeholders have different abilities and preferences which affect the chosen course of action. Agents must therefore negotiate to form teams that are both fairly aligned with individual interests and capable of achieving the task at hand.

This problem can formalized as a *team-formation negotiation* task as follows (Kraus, 1997; Shehory & Kraus, 1998). By definition, no single agent can perform the task on their own, but there may be several teams of agents who are capable of doing so, so each agent must decide who to collaborate with. The reward for accomplishing the task is awarded to the first team that solves it. Hence, agents need to interact with one another to simultaneously form a team and agree on how to share the joint reward. To solve this abstract problem, one must provide a concrete environment where agents can negotiate and reach an agreement; We must specify a *negotiation protocol* that encodes the allowed negotiation actions and determines the agreement reached (Rosenschein & Zlotkin, 1994).

Team-formation negotiation tasks are natural objects of study in game theory. More precisely, *cooperative* game theory focuses on interactions between agents who form teams and make enforceable agreements about outcomes (Brandenburger, 2007; Chalkiadakis et al., 2011). [1] *Weighted voting games* are an archetypal problem, in which every agent has a weight and a team of agents is successful if the sum of the weights of its participants exceeds a fixed threshold (Banzhaf III, 1964; Holler, 1982). Weighted voting games also offer a simple model of coalition formation in legislative bodies (Leech, 2002; Felsenthal et al., 1998). Cooperative game theory seeks to predict the agreements negotiated by agents in such settings, proposing several solution concepts. Some solutions, such as the core and nucleolus (Schmeidler, 1969), have focused on identifying stable agreements. Other solutions, known as *power indices*, have tried to measure the objective negotiation position

---

[1]This setting is akin to *ad hoc teamwork* (Stone et al., 2010), except it has an initial phase where agents negotiate before solving the problem together, and make binding agreements about sharing a joint reward.

of agents, quantifying their relative ability to affect the outcome of the game, or the *fair* share of the joint reward they should receive (Chalkiadakis et al., 2011). The most prominent of these is the Shapley value (Shapley, 1953b) which has been widely studied for weighted voting games (Shapley & Shubik, 1954; Straffin, 1988). In particular, it has been used to estimate political power (Leech, 2002; Felsenthal et al., 1998). In Appendix A we provide a detailed motivating example, showing how the Shapley value fairly measures power in such settings.

There remains a pragmatic question for the design of multi-agent systems. How should one construct a negotiating agent that maximizes the reward obtained? Many researchers have borrowed ideas from cooperative game theory to hand-craft bots (Zlotkin & Rosenschein, 1989; Aknine et al., 2004; Ito et al., 2007), often requiring additional human data (Oliver, 1996; Lin & Kraus, 2010). Such bots are tailored to specific negotiation protocols, so modifying the protocol or switching to a different protocol requires manually re-writing the bot (Jennings et al., 2001; An et al., 2010). As a result, algorithms based purely on cooperative game theory are neither generally applicable nor scalable.

Moreover, negotiation and team formation in the real world is significantly more complex than in the game theoretic setting, for several reasons: (1) negotiation protocols can be arbitrarily complicated and are rarely fixed; (2) enacting a negotiation requires a temporally extended policy; (3) the idiosyncrasies of the environment affect the negotiation mechanics; (4) Players must make decisions based on incomplete information about others' policies.

We propose multi-agent reinforcement learning as an alternative paradigm which may be applied to arbitrary negotiation protocols in complex environments. Here, individual agents must learn how to solve team formation tasks based on their experiences interacting with others, rather than via hand-crafted algorithms. Our RL approach is automatically applicable to Markov games (Shapley, 1953a; Littman, 1994), which are temporally and spatially extended, similar to recent work in the non-cooperative case (Leibo et al., 2017; Lerer & Peysakhovich, 2017; Foerster et al., 2017). In contrast to earlier work on multi-agent RL in non-cooperative games, the key novelty of our work is comparing the behaviour of negotiating RL agents with solutions from *cooperative game theory*.

Some previous work in multi-agent (deep) reinforcement learning for negotiation has cast the problem as one of communication, rather than team formation (e.g. Georgila & Traum (2011); Lewis et al. (2017); Cao et al. (2018b)). In particular, the environments considered involved only two agents, sidestepping the issue of coalition selection. Closer to our perspective is the work of Chalkiadakis & Boutilier (2004); Matthews et al. (2012), which propose a Bayesian reinforcement learning framework for team formation. However, they do not consider spatially extended environments and the computational cost of the Bayesian calculation is significant.

We evaluate our approach on a team formation negotiation task using a direct negotiation protocol, showing that agents trained via independent reinforcement learning outperform hand-crafted bots based on game-theoretic principles. We analyze the reward distribution, showing a high correspondence with the Shapley value solution from cooperative game theory. We show that the slight deviation is not due to lack of neural network capacity by training a similar-sized *supervised* model to predict the Shapley value. We also introduce a more complicated spatial grid-world environment in which agents must move around to form teams. We show that the correspondence with the Shapley value persists in this case, and investigate how spatial perturbations influence agents' rewards.

## 2 DEFINITIONS

**Cooperative Games**   We provide definitions from cooperative game theory that our analysis uses (for a full review see Osborne & Rubinstein (1994) and Chalkiadakis et al. (2011)). A (transferable-utility) *cooperative game* consists of a set $A$ of $n$ agents, and a *characteristic function* $v : P(A) \rightarrow \mathbb{R}$ mapping any subset $C \subseteq A$ of agents to a real value, reflecting the total utility that these agents can achieve when working together. We refer to a subset of agents $C \subseteq A$ as a *team* or *coalition*. A *simple cooperative game* has $v$ take values in $\{0, 1\}$, where $v(C) = 1$ iff $C$ can achieve the task, modelling the situation where only certain subsets are viable teams. In such games we refer to a team $X$ with $v(X) = 1$ as a *winning team*, and a team $Y$ with $v(Y) = 0$ as a *losing team*. Given a winning team $C$ and an agent $a \in C$, we say that $a$ is *pivotal* in $C$ if $v(C \setminus \{a\}) = 0$, i.e. $a$'s removal from $C$ turns it from winning to losing. An agent is pivotal in a permutation $\pi$ of agents if they are pivotal in the set of agents occurring before them in the permutation union themselves. Formally, let $S_\pi(i) = \{j | \pi(j) < \pi(i)\}$. Agent $i$ is pivotal in $\pi$ if they are pivotal for the set $S_\pi(i) \cup \{i\}$.

**Shapley Value** The Shapley value characterizes fair agreements in cooperative games (Shapley, 1953b). It is the only solution concept fulfilling important fairness axioms, and is therefore an important quantity in cooperative games (Dubey, 1975; Dubey & Shapley, 1979; Straffin, 1988). The Shapley value measures the proportion of all permutations in which an agent is pivotal, and is given by the vector $\phi(v) = (\phi_1(v), \phi_2(v), \ldots, \phi_n(v))$ where

$$\phi_i(v) = \frac{1}{n!} \sum_{\pi \in \Pi} \left[ v(S_\pi(i) \cup \{i\}) - v(S_\pi(i)) \right]. \tag{1}$$

**Weighted voting games** A *weighted voting game* $[w_1, w_2, \ldots, w_n; q]$ is a simple cooperative game described by a vector of weights $(w_1, w_2, \ldots, w_n)$ and a quota (threshold) $q$. A coalition $C$ wins iff its total weight (the sum of the weight of its participants) meets or exceeds the quota. Formally $v(C) = 1$ iff $\sum_{i \in C} w_i \geq q$. By abuse of notation, we identify the game description with the characteristic function, writing $v = [w_1, w_2, \ldots, w_n; q]$. The Shapley value of weighted voting games has been used to analyze political power in legislative bodies, and for such settings it is referred to as the Shapley-Shubik power index (Shapley & Shubik, 1954).

**Multi-agent Reinforcement Learning** An $n$-player *Markov game* specifies how the state of the an environment changes as the result of the joint actions of $n$ individuals. The game has a finite set of states $\mathcal{S}$. The observation function $O : \mathcal{S} \times \{1, \ldots, n\} \to \mathbb{R}^d$ specifies each player's $d$-dimensional view of the state space. We write $\mathcal{O}^i = \{o^i \mid s \in \mathcal{S}, o^i = O(s, i)\}$ to denote the observation space of player $i$. From each state, players take actions from the set $\mathcal{A}^1, \ldots, \mathcal{A}^n$ (one per player). The state changes as a result of the joint action $a^1, \ldots, a^n \in \mathcal{A}^1, \ldots, \mathcal{A}^n$, according to a stochastic transition function $\mathcal{T} : \mathcal{S} \times \mathcal{A}^1 \times \cdots \times \mathcal{A}^n \to \Delta(\mathcal{S})$, where $\Delta(\mathcal{S})$ denotes the set of probability distributions over $\mathcal{S}$. Each player receives an individual reward defined as $r^i : \mathcal{S} \times \mathcal{A}^1 \times \cdots \times \mathcal{A}^n \to \mathbb{R}$ for player $i$. In *reinforcement learning* each agent independently learns, through its own experience, a behavior policy $\pi^i : \mathcal{O}^i \to \Delta(\mathcal{A}^i)$ (denoted $\pi(a^i | o^i)$) based on its observation $o^i$ and reward $r^i$. Each agent's goal is to maximize a long term $\gamma$-discounted payoff (Sutton & Barto, 2018). Agents are independent, so the learning is *decentralized* (Bernstein et al., 2002).

## 3 GAME SETTING

### 3.1 OVERVIEW

We propose a method for training agents to negotiate team formation under a diverse set of negotiation protocols. In this setting, many different combinations of agents can successfully perform a task. This is captured by an *underlying cooperative game*, given by a characteristic function $v : P(A) \to \{0, 1\}$ (which maps viable teams $C \subseteq A$ to the value 1, and non-viable teams to 0). When a viable team is formed, it obtains a reward $r$, to be shared between the team's individual members. The outcome of the team formation process is an agreement between agents of a viable team regarding how they share the reward $r$ (with each agent trying to maximize its share). Cooperative game theory can characterize how agents would share the joint reward, by applying solution concepts such as the Shapley value. However, cooperative game theory abstracts away the mechanism by which agents negotiate. To allow agents to interact, form teams and reach an agreement regarding sharing the reward, one must use a *negotiation protocol*, forming an environment with specific rules governing how agents interact; this environment consists of the actions agents can take, and their semantics, determining which team is formed, and how the reward is shared.

We examine two simple negotiation protocols, a non-spatial environment where agents take turns making offers and accepting or declining them, and a spatial environment where agents control their demanded share of the reward in a grid-world setting. Overlaying the underlying cooperative game with a specific negotiation protocol yields a *negotiation environment*; this is a Markov game, which may be analyzed by non-cooperative game theory, identifying the equilibrium strategies of agents. However, solving even the simplest such games is computationally infeasible: even the restricted case of an unrepeated two-player general-sum game is computationally hard to solve, being PPAD-complete (Chen & Deng, 2006) (see Appendix C). Instead, we propose training independent RL agents in the negotiation environment. While our methodology can be applied to any cooperative game, our experiments are based on a weighted voting game as the underlying cooperative game. We examine the relation between the negotiation outcome that RL agents arrive at and the cooperative game theoretic solution, as is illustrated in Figure 1.

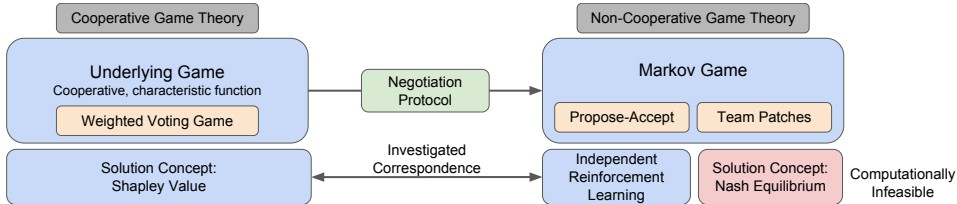

Figure 1: An overview of our methodology. We examine a team formation task defined by an underlying cooperative game. Applying different negotiation protocols to the same underlying task generates different environments (Markov games). Instead of hand-crafting negotiation bots to each such environment, we train independent RL negotiation agents. We compare the agreements RL agents arrive at to game theoretic solutions to the underlying cooperative game.

## 3.2 NEGOTIATION ENVIRONMENTS

Our negotiation environments use a weighted voting game $[w_1, \ldots, w_n; q]$ as the underlying game, offering a total reward $r \in \mathbb{N}$. Each agent $i$ is assigned a weight $w_i \in \mathbb{R}$ at the beginning of each episode, and the goal of the agents is to construct a team $C$ whose total weight $\sum_{i \in C} w_i$ exceeds a fixed quota $q$. If a successful team forms, the *entire* team is assigned the fixed reward $r$. The agents comprising the team must decide how to allocate this reward amongst themselves by agreeing on shares $\{r_i\}_{i \in C}$ such that $\sum_{i \in C} r_i = r$, with $r_i \in \mathbb{N}_0$. We say a team $C \subseteq A$ is *viable* if $\sum_{i \in C} w_i \geq q$. Though there are many viable teams, not all teams are viable (depending on the weights). Only one viable team is chosen, and non-members all get a reward of $0$. In such settings, agents face opportunities to reach agreements for forming teams, along with their share of the gains. If they do not agree, they stand the risk that some other team would form without them (in which case they get no reward); on the other hand, if they agree to a low share of the reward, they miss out on the opportunity to reach an agreement with others giving them a potentially higher share.

Our two negotiation environments have an identical underlying game, but employ different protocols for negotiating over the formed team and reward allocation. The non-spatial environment uses direct proposals regarding formed teams and reward allocations, while our spatial environment has a similar interpretation to that of recent work applying multi-agent deep reinforcement learning to non-cooperative game theory settings like spatially extended seqential social dilemmas (e.g. Leibo et al. (2017); Perolat et al. (2017); Hughes et al. (2018)).

### 3.2.1 PROPOSE-ACCEPT (NON-SPATIAL)

In the Propose-Accept environment, agents take turns in proposing an agreement and accepting or declining such proposals. The underlying game, consisting of the weights and threshold, is public knowledge, so all agents are aware of $v = [w_1, \ldots, w_n; q]$ chosen for the episode. Each turn within the episode, an agent is chosen uniformly at random to be the *proposer*. The proposer chooses a viable team and an integer allocation of the total reward between the agents. The set of actions for the proposer agent consists of all n-tuples $(r_1, r_2, \ldots, r_n)$, where $r_i \in \mathbb{N}_0$ and $\sum_{i=1}^n r_i = r$. By convention, the selected team $C$ consists of the agents with non-zero reward under this allocation; that is to say $C = \{i | r_i > 0\}$. We refer to the agents in the team chosen by the proposer as the *proposees*. Once the proposer has made a proposal, every proposee has to either *accept* or *decline* the proposal. If all proposees accept, the episode terminates, with the rewards being the ones in the proposed allocation; if one or more of the proposees decline, the entire proposal is declined. When a proposal is declined, with a fixed probability $p$ a new round begins (with another proposer chosen uniformly at random), and with probability $1 - p$ the game terminates with a reward of zero for all agents, in which case we say the episode terminated with agents *failing to reach an agreement*.

All proposals consist of a viable team and an allocation $(r_1, \ldots, r_n)$ such that $\sum_{i=1}^n r_i = r$, so the total reward all the agents obtain in each episode, is either exactly $r$ (when some proposal is accepted), or exactly zero (when agents fail to reach an agreement). Interactions in the Propose-Accept environment can be viewed as a *non-cooperative* game, however solving this game for the equilibrium behavior is intractable (see Appendix C for details).

### 3.2.2 TEAM PATCHES (SPATIAL)

We construct a spatial negotiation environment, based on the same underlying weighted voting game as in the Propose-Accept environment, called the Team Patches environment (shown in Figure 2). This is a $15 \times 15$ grid-world that agent can navigate, including several colored rectangular areas called *patches*. Agents can form a team with other agents occupying the same patch, demanding a share of the available reward to be a part of the team. Similar to Propose-Accept, the underlying game (i.e. the agent weights and the quota) are fully observable by all agents. Additionally, as the environment is spatial, agents also observe their surrounding area (see the Appendix for details).

At the start of an episode, agents are randomly initialized in the center of the map. Each step, they can move around the environment (forwards, backwards, left, right), rotate (left, right), and set their *demand* ($d_i \in \{1, 2 \ldots, r\}$), indicating the minimal share of the reward they are willing to take to join the team. To form teams, agents must move into patches, and we refer to all agents occupying the same patch as the patch's team. The team $T^j$ in patch $j$ is *viable* if the total weight of the agents in $T^j$ is greater than the threshold ($\sum_{i \in T^j} w_i \geq q$). The demands of the patch's team are *valid* if the total demanded is less than the total available ($\sum_{i \in T^j} d_i \leq r$). An agreement is reached once there is a patch whose team is both viable and with a valid reward allocation. We show an example in Figure 2(b). An episode terminates when an agreement is reached on one of the patches $j$, giving each agent $i$ in the team $T^j$ their demanded reward ($d_i$). Unlike the Propose-Accept environment, the agents don't necessarily use up all the reward available, since $\sum_i d_i < r$ is allowed. If the agents fail to reach an agreement after 100 steps, the episode is terminated, and all agents receive 0 reward.

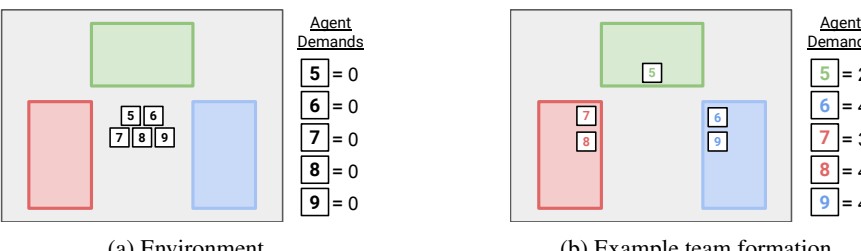

(a) Environment.          (b) Example team formation.

Figure 2: The Team Patches environment for a game with threshold $q = 15$ and a total reward of $r = 7$. Agents are represented by the squares containing their weights. In this example, we have $n = 5$ agents, with the weights $v = [5, 6, 7, 8, 9]$, and three patches where agents can form teams (red (left), green (top), and blue (right)). (a) At the start of an episode, the agents randomly begin in the center of the map with no assigned team or demands. (b) At the end of an episode, the agents have moved inside the grid-world to each of the patches. Agents in red (7 and 8) form a viable team with a valid reward allocation as their weights are above the required threshold ($7 + 8 \geq 15$) and their demands are equal to the availability ($3 + 4 \leq 7$). The team in green is not viable as the total weight is not sufficient ($5 \not\geq 15$), and the blue team has an invalid reward allocation as their demands are higher than the availability ($4 + 4 \not\leq 7$). Agents 7 and 8 receive 3 and 4 reward respectively.

### 3.3 LEARNING AGENTS

For Propose-Accept, each agent independently learns a policy using SARSA($\lambda$) (A. Rummery & Niranjan, 1994) with $\lambda = 0.1$. We use a function approximator to learn the $Q$ function. We train using the Adam optimizer (Kingma & Ba, 2014) to minimize the online temporal difference error. The function approximator is a multi-layer perceptron with 3 hidden layers, each of size 64. For Team Patches, we use an advantage actor-critic algorithm (Mnih et al., 2016) with the V-trace correction (Espeholt et al., 2018), learning from 16 parallel copies of the environment with an on-policy algorithm. The neural network uses a convolutional layer with 6 channels, kernel size 3 and stride 1, followed by a multi-layer perspective with 2 hidden layers of size 32. The policy and value function heads are linear layers. Our setup is analogous to Jaderberg et al. (2018), but using no evolution.

## 4 EXPERIMENTS

Our experiments are based on a distribution $D$ over underlying weighted voting games. We sample games from this distribution, and refer to them as the experiment's *boards*; each board consists of

its weights and threshold ($v^i = [w_1^i, \ldots, w_n^i; q^i]$). Each such board has $n = 5$ agents, a threshold of $q = 15$, and weights sampled from a Gaussian distribution $w_i \sim \mathcal{N}(6, 1)$, and exclude boards where all players have identical power. We partition the boards sampled from the distribution to to a *train set* and a *test set*. Agents are trained for $500,000$ games using train set boards and are evaluated on the test set, requiring them to learn a general strategy for negotiating under different weights, and to be have to generalize to negotiation situations they have not encountered during training. We use a train set of 150 unique boards and a test set of $k = 50$ unique boards.

## 4.1 EXPERIMENT 1: COMPARISON WITH HAND-CRAFTED BOTS

We compare the negotiation performance of RL agents trained in our framework with hand-crafted bots. While hand-crafted bots can form excellent negotiators, they can only be used for a specific negotiation protocol. For simplicity, we examine the Propose-Accept protocol where the bot faces two kinds of decisions: (1) what offer to put in as a proposer, and (2) which offers to accept as a proposee. As our baseline, we use a *weight-proportional* bot. Given a proposal, the bot uses the proportion of its weight in the proposed team as a "target" share. The more that is proposed beyond this target, the more likely it is to accept. Bots with a similar design have shown good negotiation performance against both other bots and people (Lin et al., 2008; Lin & Kraus, 2010; Baarslag et al., 2012; Mash et al., 2017). As a proposer, bot $i$ chooses a viable team $C \ni i$ with the weight-proportional allocation $p_i = r w_i / (\sum_{i \in C} w_i)$. As a proposee, bot $i$ computes its share of the same target allocation $p_i$ and compares it with the amount $r_i$ offered to it, and accepts with probability $\sigma(c(r_i - p_i))$ where $\sigma$ is the logistic function (see the Appendix for full details of the bot). As a more sophisticated baseline, we use a *Shapley-proportional* bot, which follows the same design as the weight-proportional bot, except it sets the target allocation to be the one proportional to the Shapley values rather than the weights, i.e. $p_i = r \phi_i / (\sum_{i \in C} \phi_i)$ (where $\phi_i$ denotes the Shapley value of agent $i$). We note that as computing the Shapley values is an NP-hard problem (Elkind et al., 2009) this method is tractable for 5 agents, but does not scale to games with many agents.

We create two agent teams (called a *team pair*), one with of 5 RL agents (all-RL group), and one with of 4 RL agents and one bot (bot group). Each group is co-trained over training set boards, and evaluated for $500$ episodes on an evaluation board. We compare the amount won by the bot and RL agent with the same weight in the board. We repeat the analysis for all evaluation boards, creating $200$ team pairs for each board. In the all-RL group, each agent makes on average $0.2 \cdot r$, whereas in the one-bot group, a weight-proportional bot makes on average $0.178 \cdot r$ and a Shapley-proportional bot makes on average $0.185 \cdot r$. The difference is significant at the $p < 0.005$ level, using a Mann-Whitney-Wilcoxon test. Also, against a random bot that selects valid actions uniformly we get more significant results ($p < 0.001$). One may construct more sophisticated bots or tune their parameters to improve performance, but these results indicate that RL agents can outperform sensible hand-crafted bots. The above results are robust to the hyper-parameter choices for our RL agents; RL agents outperform the bots, at the same statistical significance, even when perturbing the learning rate, hidden layer sizes and the $\lambda$ parameters by $\pm 25\%$.

## 4.2 EXPERIMENT 2: CONSISTENCY WITH GAME-THEORETIC PREDICTIONS

In this experiment, we investigate whether our trained agents negotiate in a way that is consistent with solution concepts from cooperative game theory, focusing on the Shapley value. As the Shapley value was proposed as a "fair" solution, one can view this as studying whether independent-RL agents are likely share the joint gains from cooperation in a fair manner.

To investigate the negotiation policies of our RL agents, we train and evaluate them on our two environments using the same set of $k = 20$ boards. We denote the average amounts won by weight $i$ on board $j$ as $s_i^j$, and refer to these as *empirical agent rewards*.[2] Given the game $v^j$ we can compute the Shapley values, denoted as $\phi(v^j) = (\phi_1^j, \ldots, \phi_n^j)$. We view $\phi_i^j$ as the game theoretic prediction of the fair share the agent with weight $w_i^j$ should get in a negotiation in the game $v^j$. We apply the analysis to each of our $k = 20$ boards and obtain $k \cdot n$ pairs $\{(\phi_i^j, s_i^j)\}_{i \in [n], j \in [k]}$ of Shapley value predictions and empirical agent rewards. In Figure 3, we scatter and density-plot these pairs.

---

[2]For Propose-Accept, we average over 200 independent runs of 5000 episodes each, and for Team Patches we average over 100 independent runs of 1000 episodes each.

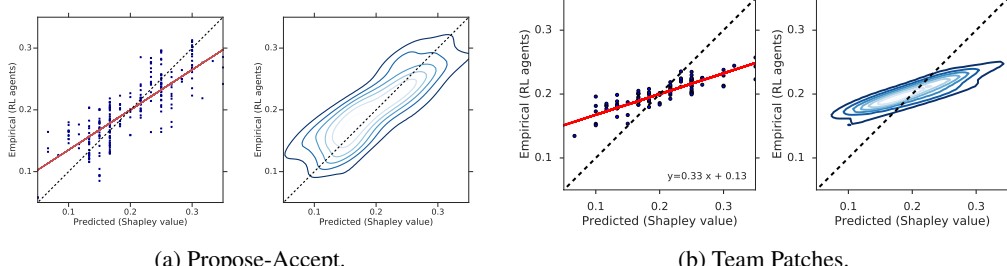

(a) Propose-Accept.  (b) Team Patches.

Figure 3: Share of reward received by agents in both environments. Shapley points (left) and corresponding density-estimation contours (right). The x-axis is the fair share prediction using the Shapley value, and the y-axis is the empirical reward share of the RL co-trained agents. We include $y = x$ (black dashed) to show the reward our agents would receive if they exactly matched the predictions, as well as a trend line (in red).

Our results indicate a strong correlation between the Shapley value based predictions and the empirical gains of our independent RL trained negotiator agents. We see that the majority of the mass occurs in close proximity to the line $y = x$, indicating a good fit between the game theoretic prediction and the point to which our RL agents converge. RL agents exhibit a stronger deviation from the game theoretic "fair" reward on the boards where there are players with significantly more power or significantly less power than others, i.e. boards where there is a high inequality in Shapley values. In these boards, "strong" players (according to the Shapley value) do get a higher share than "weak" players, but not to the extent predicted; for these rare boards, the empirical allocation of reward achieved by the RL-agents is more equal than predicted by the Shapley value. Further evaluation in Appendix E shows that having a lower weight variance in the board distribution leads to a higher correspondence between the Shapley value and the empirical gains of RL agents.

In summary, our independent RL negotiation agents empirically achieve rewards that are consistent with the outcomes predicted by the Shapley value from cooperative game theory. Notably, this consistency extends to our spatial negotiation environment even with its different negotiation protocol and spatial interactions. In Section 4.4 we investigate potential reasons for these variations.

### 4.3 EXPERIMENT 3: INFLUENCE OF SPATIAL PERTURBATIONS

As the Team Patches environment is a spatial one, we can examine how spatial aspects, that are abstracted away in the underlying cooperative game, affect negotiation outcomes. How does changing the spatial structure of the environment influence the "power" of agents? Can we reduce the share of the total reward received by the "most powerful" agent (with the highest weight)?

We consider two patches and vary the starting position of the maximal weight agent. We measure how the agent's share of the total reward changes as we change its starting position from directly touching a patch to being 10 steps away from the average patch location. Every other agent always starts 3 steps away from the nearest patch. We visualize the results in Figure 4 (more details are given in Appendix D). When the highest-weight agent is closer to the patch than the other agents, it can start negotiating a team earlier, and due to its high weight it can demand a high reward. However, as we move the agent further away, it takes longer for the agent to reach a patch, so the other (weaker) agents have more time to form a team; thus, its share of the reward significantly drops.

### 4.4 EXPERIMENT 4: WHY DO RL AGENTS DEVIATE FROM THE SHAPLEY VALUE?

While for most boards our RL agents do indeed converge to an outcome that approximates the Shapley value, there is a deviation on boards where some players have particularly strong or particularly weak negotiation positions (Shapley values). Multiple factors may contribute to this. One potential factor relates to the *representational capacity* of the RL agents; computing the Shapley value in a weighted voting game is an NP-hard problem (Elkind et al., 2009), so perhaps such a function cannot be easily induced with a neural network. Even if a neural network can compute the negotiation position in the underlying weighted voting game, we may still have an optimization error; RL agents optimize for their individual reward when negotiating under a specific protocol, and it might

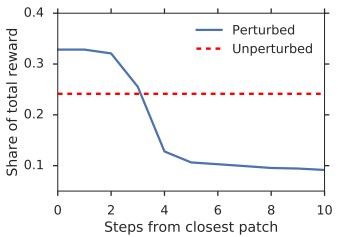
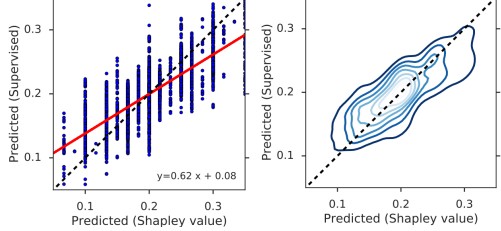

Figure 4: Influence of spatial changes (perturbed, blue) versus no changes (unperturbed, red) on the highest-weighted agent's reward.

Figure 5: Prediction of the Shapley values from the game weights and threshold using supervised learning.

be difficult to express a policy exactly fitting the Shapley value in the negotiation protocols we've selected. Finally, the deviation might be caused by the *learning dynamics* of independent RL agents; the highly non-stationary environment induced by agents learning at the same time may lead agents to agreements deviating from the cooperative game theoretic predictions.

We show that a neural network can approximate the Shapley value under supervised learning (i.e. when it does not need to consider the environment resulting from a specific negotiation protocol). We train a model to take the parameters of a weighted voting game (i.e. agent weights and the threshold), and output the Shapley values of each player. We generate $3,000$ *unique* boards from the same distribution with Gaussian weights used for the experiments in Section 4.2, and apply a train/test partition (80%/20% of the data). We then fit a simple 3-layer MLP with 20 hidden units (smaller than used in the RL agents), minimizing the mean-squared error loss between the model's predictions and the Shapley values. The results of this are shown in Figure 5.

We find that the MLP generalizes to unseen boards, and better matches the game theoretic predictions of the Shapley value. We have also conducted another experiment where we let RL agents observe not only their weights, but also their Shapley values in the underlying game. However, even in this setting we note a discrepancy between the Shapley value and the outcomes that RL agents arrive at (see Appendix F for full details). We thus believe that the deviation is not caused by the RL agents' inability to determine their relative negotiation power; The RL agents either do not manage to find a strong policy under the specific negotiation protocol, or converge on an outcome different from the Shapley value due to having a non-stationary environment with independent learners.

## 5    CONCLUSIONS AND FUTURE WORK

Team formation is an important problem for multi-agent systems, since many real-world tasks are impossible without the cooperation and coordination of multiple agents. Our contributions are as follows: (1) we introduced a scalable method for team-formation negotiation based on deep reinforcement learning which generalizes to new negotiation protocols and does not require human data, (2) we showed that negotiator agents derived by this method outperform simple hand-crafted bots, and produce results consistent with cooperative game theory, 3) we applied our method to spatially and temporally extended team-formation negotiation environments, where solving for the equilibrium behavior is hard, and (4) we showed that our method makes sensible predictions about the effect of spacial changes on agent behavioral and negotiation outcomes.

This work opens up a new avenue of research applying deep learning to team-formation negotiation tasks. In particular, it would be interesting to analyze how team formation dynamics affect emergent language in reinforcement learning agents, naturally extending the work of (Cao et al., 2018a) and (Lazaridou et al., 2016). Indeed, it has been suggested that the human ability to negotiate and form teams was critical in the evolution of language (Thomas & Kirby, 2018). One might also consider creating tasks that interpolate between the fully cooperative game-theoretic setting and the purely non-cooperative one. Fundamentally, binding contracts are managed by dynamic institutions, whose behavior is also determined by learning. In principle, we could extend our method to this hierarchical case, perhaps along the lines of (Greif, 2006). Perhaps such an analysis would even have implications for the social and political sciences.

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

## A    MOTIVATING EXAMPLES FOR APPLICATIONS OF THE SHAPLEY VALUE

Our results compare the agreements of RL agents to the Shapley value in a weighted voting game setting. To illustrate and motivate the importance of the Shapley value and other power indices, let us discuss a few example domains. Consider multiple providers such as travel agencies or airline carriers which can allow a person to travel between various destinations, and a client who is willing to pay a certain amount to get to a desired destination; while there is no direct flight to the client's destination, there are multiple routing options, using different carrier combinations. How should the carriers share the customer's payment? Similarly, consider a manufacturing scenario where multiple companies can provide subsets of components required to manufacture an end product, and where each company has only some of the components. How should they share the profit from selling the end product?

Both the above scenarios can be captured as a cooperative game (and one can follow our paradigm to train RL agents to negotiate in such domains). Solution concepts from cooperative game theory can analyze such domain, and make predictions regarding how agents are likely to share the joint gains. However, the most prominent example for applying cooperative game theory to analyze the negotiation power of agents originates from measuring political power in decision making bodies (Straffin, 1988; Shapley & Shubik, 1954; Felsenthal et al., 1998). We illustrate how power indices, and the Shapley value in particular, formalize power in a way that depends on the possible teams that can form, rather than the direct parameters of the game (such as the weights).

Consider the formation of a coalition government following an election, where a set of political parties each obtained a certain number of parliament seats in an election, and where a quota of the majority of the seats is required to form a ruling government. If no single party has won the majority of the votes on its own, multiple parties would have to join so as to form a ruling coalition government. Parties in this setting would have to negotiate to form a ruling government. This is rare in the UK, and has never happened in the USA, but is the typical situation in many other countries.

For intuition, consider a parliament of 100 seats, two big parties with 49 seats each and a small party with 2 seats; a majority requires 50 seats (half of the total number of seats), so no party can form a government on its own. While each big party has far more seats than the small party, any team of two or more parties has the required majority of seats. Under such a framing of the domain, any team of at least two out the three parties is sufficient for completing the task of forming a government, and due to this symmetry between agents one may claim they all have equal power. In other words, what matters in making a party powerful is not its number of parliament seats, but rather the number of opportunities it has to form teams. Intuitively, one would say that the small party has a strong negotiation position. It could, for instance, demand control of a dis-proportionally high part of the budget (i.e. although it only has 2% of the seats, it is likely to get control of a much larger share of a the budget).

We note that for the above example, the Shapley value would give each party the same share. In fact, the Shapley value as a function does not even take the weights or threshold as input, but rather the characteristic function of the game.

We say that agents $i, j$ are *equivalent* in a game with characteristic function $v$ if for any coalition $C$ that contains neither $i$ nor $j$ (i.e. any $C$ such that $i \notin C$ and $j \notin C$) we can add either $i$ or $j$ to the coalition and obtain the same value (i.e. $v(C \cup \{i\}) = v(C \cup \{j\})$). We note that in the above example, all agents are equivalent, and that the Shapley value allocates them the same share. This is not a coincidence — one of the fairness axioms characterizing the Shapley value is that equivalent agents get the same share (Shapley, 1953b; Dubey, 1975). Indeed, the Shapley value is the only index fulfilling a small set of fairness axioms such as this one.

Two additional axioms that fully characterize the Shapley value, in addition to the above equivalence axiom, are that of giving null players no reward, and the additivity axiom. A null player is one which adds no value to any coalition, and the null player axioms states that null players would get no reward; the additivity axioms relates to the sum of multiple games, stating that the value allocated to any player in the sum-game would be the sum of the values in the individual composing games. For a more detailed discussion of such axioms, see textbooks and papers on the axiomatic characterization of the Shapley value (Straffin, 1988; Chalkiadakis et al., 2011).

# B  BEATING BASELINE BOTS: FULL EXPERIMENT DETAILS

Section 4.1 describes our experiment comparing the negotiation performance of RL agents and hand-crafted bots. We now provide a more detailed discussion of this experiment. We first note that although the negotiation protocols differ across environments, the essence of the decisions agent face is similar. When agents fail to reach an agreement they obtain no reward, so agents who almost never reach an agreement would be bad negotiators. On the other hand, reaching agreement easily by itself does not make an agent a strong negotiator. For instance, an agent who accepts any offer is likely to reach agreement easily but perform poorly, as they would take even very low offers (even when they have a high relative weight and thus a strong negotiation position).

Although the essence of the task is similar across environments, the observations, action spaces and semantics of interaction differ considerably across negotiation protocols. The key advantage of using RL to train negotiation agents is not having to hand-craft rules relating to the specifics of the negotiation environment. Indeed, as hand-crafting a bot is a time consuming process, we focused on the simpler propose-accept environment; creating a hand-crafted bot for the team patches environment is more difficult, as there are many more decisions to be made (how to find a good patch, which share to demand, how to respond if the total reward demanded is too high, etc.)

Our simplest baseline is a *random bot*. As a proposer, it selects an allowed proposal uniformly at random from the set of all such proposals. As a proposee it accepts with probability $\frac{1}{2}$ and rejects with probability $\frac{1}{2}$.

A more sensible basedline is the *weight-proportional bot*, which is an adaptation of previously proposed negotiation agents to the propose-accept protocol (Lin et al., 2008; Lin & Kraus, 2010; Baarslag et al., 2012; Mash et al., 2017) (the propose-accept protocol itself bears some similarity to the negotiation protocols used in such earlier work on negotiation agents). An even stronger baseline is the *Shapley-proportional bot*, which is similar to the weight-proportional bot, but computes the Shapley values of all the agents (which is computationally tractable only for games with few agents).

Section 4.1 provides a short description of the actions taken by the weight-proportional and Shapley proportional bots. We provide a longer description in this appendix, trying to elaborate on the intuition and principles behind the design of these bots. As a proposer, the weight-proportional bot randomly chooses a viable team $C$ which contains itself from the set of all such teams. It then proposes an allocation of the reward that is proportional to each the team agents' weights. Formally, given a board $[(w_1, \ldots, w_n); q]$, we denote the total weight of a team $C$ by $w(C) = \sum_{i \in C} w_i$. For the team it chose, the bot uses the target allocation $p_i = \frac{w_i}{w(C)} \cdot r$ where $r$ is the fixed total reward. [3]

As a proposee, the weight-proportional bot computes its share of the same target allocation $p_i = \frac{w_i}{w(C)} \cdot r$ in the proposed team $C$, and compares it with the amount $r_i$ offered to it in the proposal. We denote by $g_i = r_i - p_i$ the amount by which the offer $r_i$ exceeds $p_i$, the bot's expectations under the weight-proportional allocation. A high positive amount $g_i$ indicates that the bot is offered much more than it believes it deserves according to the weight-proportional allocation, while a negative amount indicates it is offered less than it thinks it deserves (when offered exactly the weight-proportional share we have $g_i = 0$). The probability of the bot accepting the offer is $\sigma(c \cdot g_i)$ where $\sigma$ denotes the logistic function $\sigma(x) = \frac{1}{1+e^{-x}}$, and where $c = 5$ is a constant correcting the fact that $g_i$ only ranges between $-1$ and $+1$ rather than between $-\infty$ and $+\infty$. Thus the bot accepts a "fair" offer (according to the weight-proportional target) with probability of $\frac{1}{2}$, and the probability convexly increases as $g_i$ increases (and decreases as $g_i$ decreases).

The more sophisticated *Shapley-proportional* bot follows the same design as the weight-proportional bot, except it sets the target allocation to be the one proportional to the Shapley values rather than the weights, i.e. $p_i = r\phi_i/(\sum_{i \in C} \phi_i)$ (where $\phi_i$ denotes the Shapley value of agent $i$).

As discussed in Section 4.1, we compare the performance of the bot and RL trained agents by creating many pairs of groups. We examine each of the evaluation boards (sampled from the distribution of Gaussian weigh boards). For each evaluation board, we create $t = 200$ *pairs* of agent groups, where in each pair we have one group of $n = 5$ independent RL agents (the *all-RL group*), and one set consisting of 1 bot and 4 independent RL agents (the *single-bot group*). Each group is co-trained

---

[3]Our framework's action space includes only integral reward allocations, so the bot proposes chooses the integral reward allocation that minimizes the $L_1$ distance to the target allocation.

for $m = 500,000$ episodes, where during training each episode uses a different board from the train set. During the evaluation step, we let each agent group play $5,000$ games on each evaluation board in the test set. We repeat the same analysis, each time allowing the bot to play a different weight in the board (i.e. letting the bot be the agent in the first position, in the second position and so on).

We investigate the performance of the RL agent from the all-RL group and the bot from the single-bot group, when playing the same weight on the same evaluation board. We average the fraction of the total fixed reward achieved by the RL-agent and bot over the $t = 200$ group pairs and over the $5000$ episodes played with each evaluation board, and examine the difference $d$ between the amount won by the RL-agent and the amount won by the bot. A positive value for $d$ indicates the RL-agent has a better performance than the bot, and a negative number shows a higher performance for the bot.

On average (across boards and episodes played on each evaluation board), the RL-agent outperforms the weight-proportional bot, achieving $0.025 \cdot r$ more of the total reward. In the all-RL group, each agent makes on average $0.2 \cdot r$, whereas in the one-bot group the bot makes on average $0.178 \cdot r$. In other words, the RL-agent obtains 10% more reward than the bot. We performed a Mann-Whitney-Wilcoxon test, which shows the difference is significant at the $p < 0.005$ level. [4] The results are similar for the Shapley-proportional bot, although the performance gap is smaller, with the Shapley-proportional bot making on average $0.185 \cdot r$ (the RL agent outperforms the Shapley-proportional bot at the same statistical significance level as for the weight-proportional bot). Unsurprisingly, performing a similar analysis for the random bot, that selects valid actions uniformly at random, we get even more significant results ($p < 0.001$).

We have performed an additional experiment, where we train the RL-agents in the one-bot group with a weight-proportional bot, but use a Shapley-proportional bot during their evaluation. In other words, their adapt their policy for one type of bot, but are measured against another type of bot. This hindrance sightly lowers the average reward share of RL agents, allowing the bot to gain a slightly larger reward share of $0.188 \cdot r$. However, even in this case the RL-agents outperform the bot ($p < 0.005$).

We note that our results indicate that RL agents can outperform *some simple heuristic* hand-crafted bots. This result certainly does not mean that it is impossible to create well-crafted bots, tailored to a specific negotiation protocol, that would out-negotiate RL agents. For instance, even for the weight-proportional and Shapley-proportional bots we can tune the distribution parameter $c$ discussed above, and possible improve the negotiation performance. We view this analysis as an indication to RL-agents make at least somewhat sensible negotiation decisions. The key advantage of the methodology we propose is its generality and ability to tackle new negotiation protocols, without relying on hand-crafted solutions. In other words, our technique offers a way of automatically constructing at least reasonable negotiators, without requiring fitting a bot to a given negotiation protocol.

## C   ON THE COMPLEXITY OF COMPUTING NASH EQUILIBRIA IN THE NEGOTIATION ENVIRONMENTS

We used the Shapley value from *cooperative* game theory to examine likely outcomes in the underlying weighted voting game. The Propose-Accept environment and the Team Patches environment are both based on this underlying game, but define the possible actions agents can take and the outcomes for various action profiles (i.e. they employ a specific negotiation protocol). Thus, they can be viewed as a *non-cooperative* (Markov) game. For such setting, one can also examine solution concepts from *non-cooperative* game theory, such as the Nash equilibrium (Nash, 1950).

The resulting Markov game is not computationally tractable to solve for the Nash equilibrium, as it is an infinitely-repeated $n$-player general-sum extensive-form game. This is among the hardest class of games to compute an equilibrium for. Even the restricted case of an unrepeated two-player general-sum game is "hopelessly impractical to solve exactly" (Shoham & Leyton-Brown, 2009, Section 4.3), being PPAD-complete (Chen & Deng, 2006). We thus choose to apply cooperative game theoretic solutions, namely the Shapley value.

---

[4]This is a non-parametric test similar to Student's T-test, but resistant to deviations from a Gaussian distribution.

## D  TEAM PATCHES ENVIRONMENT

Our spatial Team Patches environment is a $15 \times 15$ grid-world where each entity has an assigned color. The agents observe this world from their own perspective, as shown in Figure 6(a), and also observe the weights and demands of every other agent, as well as their own index.

In Experiment 3 (Section 4.3) we change this world in two ways:

1. We set the total number of patches in the world to 2 (red and blue).
2. We modify the starting position of the agent with the highest weight, investigating how this spatial perturbation influences their share of the total reward.

In Figure 6(b) we visualize this with a full view of the environment where the white agent is moved between squares 0 and 10, corresponding to being $N$ steps away from the nearest patch in $L^1$ distance.

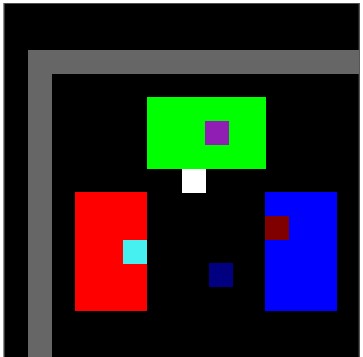
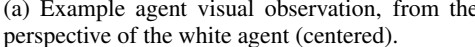
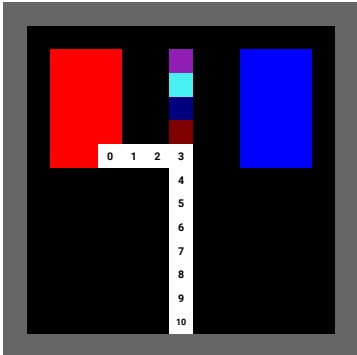

(a) Example agent visual observation, from the perspective of the white agent (centered).

(b) Visualization at the start of an episode for the spatial perturbations experiment.

Figure 6: Team Patches – (a) Example RGB observation an agent receives in an episode. The white square is the observing agent (centered). Other agents are observed as maroon, cyan, purple, and navy, patches as red, green, and blue, and the environment's border is observed as gray. In addition to these observations, the agent also receives its index as a one-hot vector of size number of players, the weights of all agents, and their current demands. (b) Visualization of spatial perturbations where the agent with the highest weight is initialized at 0 to $N$ squares from the nearest patch.

## E  THE IMPACT OF WEIGHT AND POWER INEQUALITY ON THE CORRESPONDANCE WITH THE SHAPLEY VALUE

Section 4.2 discussed the correspondence between the outcomes achived by RL agents and the Shapley value from cooperative game theory, indicating that deviations occur mostly in boards where some agents have a particularly weak or particularly strong negotiation position, as measured by the Shapley value. The board distribution $D$ that we have there used ruled-out boards where all agents have identical weights (resulting in identical power).

To investigate the impact of the weight and power inequality on the correspondence with the Shapley value, we examined an alternative board distribution $D'$, which did not rule-out equal-weight boards (the remaining weight sampling procedure was identical to the original distribution $D$). We generated a new train set and test set of boards from $D'$. The sampled weights under $D$ had a standard deviation of $STD_D = 1.65$ whereas the standard deviation of weights under $D'$ was $STD_{D'} = 1.1$. We repeated the analysis of Section 4.2 for these reduced variance boards, resulting in Figure 7.

Figure 7 shows that the mass is much more concentrated around the area with equal Shapley values (points with $x \approx 0.2$), where the outcome achieved by RL agents is also very close to the same value (i.e. $y \approx 0.2$). As the Figure shows, this results in having a stronger correspondence between the outcome obtained by RL agents and the Shapley value. In other words, high inequality in agent power (negotiation position) results in a larger discrepancy between the Shapley value and outcomes that RL agents arrive at.

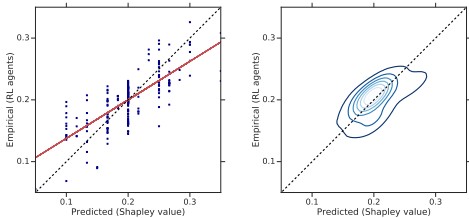

Figure 7: Shapley Correspondance on Boards With Reduced Variance

## F    LETTING AGENTS OBSERVE THEIR NEGOTIATION POSITION

In Section 4.4 we discuss potential reasons for the empirical gains of our RL agents deviating from the Shapley value. The analysis there shows that given a direct supervision signal (boards labeled with the Shapley values), a small neural net can approximate the Shapley value well. Our RL agents have a more challenging task for two reasons: (1) They have to take into account not only their negotiating position but also protocol details. (2) Their RL supervision is weaker: they only know how successful a whole **sequence** of actions was, and not the "correct" action they should have taken at every timestep.

Our conclusion from Figure 5 is that at least the basic supervised learning task can be accomplished with a small neural network, so the agents network has the capacity required to estimate their raw negotiating power, abstracting away protocol details. We now show that even when given the Shapley values in the underlying game, RL agents reach outcomes that may deviate from the Shapley value. We use an experimental setup that is identical to that used to produce Figure 3a, except we provide the Shapley values of all the agents as a part of the observation (in every state). The outcome of the experiment is shown in Figure 8.

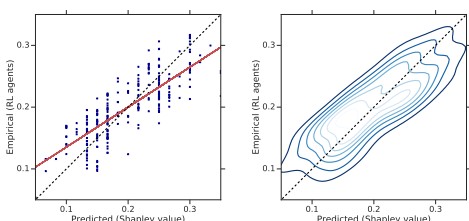

Figure 8: Shapley Aware Propose Accept

Figure 8 shows the same pattern as Figure 3a. The deviation from game theoretic predictions thus stems not from being unable to identify the "high-level" negotiation position of agents, but rather from the RL procedure we applied; RL agents attempt to maximize their own share of the gains (rather than for obtaining their "fair" share of the gains, as captured by the Shapley value), and are forced to deal with a the need to find strong *policies* taking into account the specific negotiation protocol (under a non-stationary environment, where their peers constantly adapt their behavior).

