# OpenReview forum: "Negotiating Team Formation Using Deep Reinforcement Learning"
_ICLR.cc/2019/Conference_

### Official Review · AnonReviewer3 · 2018-11-03
**Interesting exploration into RL for negotiation in coalition games**

**Rating:** 5
**Confidence:** 3

**Review:**

This paper develops a reinforcement learning approach for negotiating coalitions in cooperative game theory settings.  The authors evaluate their approach on two games against optimal solutions given by the Shapley value.

The work builds upon a substantial and growing literature on reinforcement learning for multiagent competitive and cooperative games. The most novel component of the work is a focus on the process of negotiation within cooperative coalition games. The two game environments studied examine a "propose-accept" negotiation process and a spatial negotiation process.

The main contribution of the work is the introduction of a reinforcement learning approach for negotiation that can be used in cases where unlimited training simulations are available.  This approach is a fairly straightforward application of RL to coalition games, but could be of interest to researchers studying negotiation or multiagent reinforcement learning, and the authors demonstrate the success of RL compared to a normative standard.

My primary concerns are:
- The authors advertise the work as requiring no assumptions about the specific negotiation protocol, but the learning algorithms used are different in the two cases studied, so the approach does require fine-tuning to particular cases.
- Maybe I missed it, but how many training games are required?
- In what real applications do we expect this learning algorithm to be useful?
- The experiments where the RL agents are matched against bots include training against those specific bot types. How does the trained algorithm perform when matched against agents using rules outside its training set?
- Since the Shapley value is easily computable in both cases studied.  If the bots are all being trained together, why wouldn't the bots just use that to achieve the optimal solution?
- Why are only 20 game boards used, with the same boards used for training and testing?  How do the algorithms perform on boards outside the training set?

Overall, the paper is somewhat interesting and relatively technically sound, but the contribution seems marginal.

---

> ### Author Response · Authors · 2018-11-07
> **Our results are robust to learner hyper-parameters; We'll evaluate over previously unobserved boards.**
>
> Indeed, our focus is on using multiagent reinforcement learning to train agents in negotiation, rather than using hand crafted bots tailored to a specific negotiation protocol or interaction rules.
>
> Indeed, there are many RL algorithms that can be used, each having multiple hyper-parameters (neural network architecture, learning rates, optimizer and loss, eligibility traces configuration etc.). Clearly, any application of deep RL requires setting / tuning such parameters. When comparing to the Bot baselines we found that the RL-agents beat the bots across many such settings (i.e. the success in negotiation is robust to the choice of learning algorithm and hyper-parameter settings); even a shallow function approximator (with a small hidden layer size) is sufficient to beat the baseline bot. Similarly, the correspondence with the Shapley value holds under many learner configurations. We will add a discussion of this to the paper, as well as an analysis showing such robustness. As discussed in the paper, we emphasize that the main advantage of our approach is achieving a reasonably strong negotiator without having to hand craft a bot. We believe that tuning hyper-parameters for an RL algorithm requires considerably less work than writing a full fledged bot for a negotiation protocol, which must take into account not only the negotiation position of the agents, but also nuances regarding the interaction protocol.
>
> The number of training steps for the analysis was 500,000. The full details are in the appendix (page 14), but indeed this detail belongs in the main text - we will move it there.
>
> As for potential applications of this work, consider the following motivating example: multiple providers (travel agencies / carriers) can allow a person to travel between various destinations, and a client is willing to pay a certain amount to get to a desired destination (while there is no direct flight, there are multiple routing options, using different carrier combinations). How should the carriers share the customer's payment? Similarly, consider a manufacturing scenario where multiple companies can provide subsets of components required to manufacture an end product, and where each company has only some of the components. How should they share the profit from selling the end product?  Both scenarios can be captured as a cooperative game, so RL agents can be used to learn to negotiate in such domains (for similar examples, see: Chalkiadakis, G., Elkind, E., & Wooldridge, M. (2011), Computational aspects of cooperative game theory). We will add a brief discussion of these motivating examples to the paper.
>
> As you point out, training the RL-agents with some bots, then testing them with other bots is likely to hinder the performance of these agents. We will carry out such an experiment, and examine the impact of "out of training set" bots. However, we must note that even a negotiation bot designer faces a similar problem: when designing a bot to have a good performance against a bots of type A, its performance may be sub-optimal against a bot of type B. As in any game theoretic setting, the outcome an agent achieves depends not only on its own policy, but also on the policy used by others.
>
> As you note, our experiments are based on settings with few agents (5 agents in a game), which makes it tractable to compute Shapley values. However, computing the Shapley value is an NP-hard problem, so approaches based on computing the Shapley value directly may not scale to games with many agents (while an RL approach does scale). As you propose, we will add an experiment comparing a bot which uses the Shapley value as the target for its share under the negotiation (the weight proportional bot is a rough approximation of a Shapley bot), as this is a strong negotiation baseline. The optimal policy for an agent to use depends on the policies used by other agents, so the Shapley bot may not be optimal against all agents (for instance, it may be too stubborn).
>
> Our analysis is based on 20 different board configurations, but as each board has 5 agents (and thus 5 weights), so there are 100 different negotiation positions each agent may have. Given a total payoff of 10, the action space of the agent is any integral partition of 10 points to 5 agents (which is 14 choose 4 , or over 1,000 different proposal actions), resulting in a huge policy space, even for the very simple propose accept environment. This seems a reasonably large space to explore. However, we can certainly increase the number of sampled board configurations, which we'll do in the revised version. We wholeheartedly agree that is important to see how agents perform in negotiation situations they have not encountered. We will examine performance against bots and Shapley correspondence on held-out boards, and will include this in the revised version - thanks for pointing this out!

---

> > ### Comment · AnonReviewer3 · 2018-11-13
> > **Good plan but requires execution to evaluate**
> >
> > I appreciate the authors' thoughtful response to my comments.
> >
> > If the authors were able to execute and report the promised new tests and experiments with positive results, I would be willing to revise my score.
> >
> > Regarding the need for 500,000 training steps. It's worth noting that this amount severely restricts the domain of application for the method. In what situations would there be the opportunity to train on that many real cases? This fact highlights the importance of checking whether the method performs well on out-of-sample situations and bots.

---

> > > ### Author Response · Authors · 2018-11-22
> > > **We revised the paper, with a train-test board partition (and more boards), a Shapley-based bot, an investigation of robustness to hyper-parameters, examination of out-of-training bots,  and more detailed motivating examples and discussion of the Shapley value**
> > >
> > > Thanks again for your feedback! As you can see, we have revised the paper:
> > >
> > > - We now partition boards to a train-set and a test set, and make sure agents generalize to previously unobserved boards (i.e. boards not encountered during training). The results regarding the bot-comparison and the Shapley correspondence still hold (see revised figures and numbers in Section 4.2)
> > > - We now consider a much larger set of train boards (150 boards) and evaluation boards (50 boards), rather than the 20 boards we had before.
> > > - We include a discussion of experiments regarding hyper-parameter settings. The results are robust for changing the learning rate, hidden layer sizes and labda (eligibility traces parameter) by 25% (and likely more, these are the settings we have evaluated).
> > > - We have added a couple of motivating examples for applications of negotiation in cooperative games, and a more detailed discussion of the axiom behind the Shapley value (see revised Appendix A)
> > > - We have added a Shapley-value based bot (similar to the weight-proportional bot, but using a target based on the Shapley value). RL agents are still competitive, even with this more sophisticated bot (which is computationally intractable for games with a much larger number of agents)
> > > - We have added an experiment regarding out-of-training bots. We train RL agents against a weight-proportional bot, and evaluate them against a Shapley-proportional bot. While this does hinder their performance a bit, they remain competitive.
> > > - We have added experiments regarding a board distribution with lower weight variance, yielding a stronger correspondence with the Shapley value (Appendix E). We also added experiments strengthening our experiment 4 (from section 4.4), further investigating the reasons behind the deviation from the Shapley value. Appendix F showing that even when RL agents observe the true Shapley values as a part of the state, they can deviate from the Shapley value.

---

### Official Review · AnonReviewer1 · 2018-11-06
**Interesting problem, more experiments would be nice**

**Rating:** 6
**Confidence:** 2

**Review:**

This is an emergency review, so apologies for the briefness.

The paper introduces an approach to learning negotiation strategies using reinforcement learning. The authors propose a new setup in which self-interested agents must cooperatively form teams to achieve a reward. They explore two ways of proposing agreements: one involving a random agent proposing an agreement symbolically, and another in which agents form teams by moving to the same location. Results show that RL-trained models outperform simple rule-based bots, and correlate with game-theoretic predictions. I think the paper is very well clearly presented, and tackles an interesting an important problem.

One issue I have is that as I understand it, the results are only reported for training games. Could the agents just be memorizing a good outcome for that specific environment, rather than actually learning to negotiate? Why not evaluate on held out games?

The experiments are pretty interesting, and I appreciated the last one showing that limitations are due to the difficulty of RL, rather than expressive power of the network. However, I think there are some other natural questions that could be explored, including: what kind of strategies are the models learning? Could we change the environment in such a way that the proposed approach is not sufficient? Is the choice of RL approach crucial, or does anything work? I think further experiments would strengthen the paper.

---

> ### Author Response · Authors · 2018-11-12
> **Thanks! We'll evaluate performance on held out boards (train/test board partition), evaluate against a Shapley bot baseline, and discuss the advantages/disadvantages of RL-agents vsersus alternatives (bots or human data).**
>
> Thank you for the helpful review, especially as an emergency reviewer!
>
> As you suggest, we will add an experiment where we perform a train / test partition of boards, and will evaluate agents on held-out boards to make sure they are not memorizing good actions for the specific training boards.
>
> Gaining more insight regarding the learned agent policy is tricky, as the policy relates to a large state space. One aspect we can examine in more depth is whether agents tend to agree quickly (e.g. the number of steps until a team is formed). We will add an experiment looking at this in more depth.
>
> As discussed in the response to other reviewers, we intend to add an experiment comparing the RL agents with a bot based directly on the Shapley value. Such a bot does not scale to many players as computing the Shapley value is an NP hard problem, but in games with 5 players which we use in our experiments it is possible to compute in reasonable time.
>
> Regarding the necessity of an RL approach: the key advantage using our RL-based approach is being able to handle diverse negotiation protocols and environments. We see two possible alternatives to RL: building a hand-crafted bot, designed for a specific negotiation protocol, or gathering data from humans who engage in negotiation and training a bot to mimic human participants.
> Both of the above alternatives are tailored to a specific negotiation protocol, and are very costly (either in gathering enough human data, or in designing and engineering the bot). Although very costly, these alternatives can achieve potentially higher quality negotiation policies. In our analysis we have noticed that for boards where some players have a very strong or very weak negotiation position, there is a more noticeable deviation from the Shapley value.
> We will add an experiment examining the impact of the weight variance (or similarly, the degree of inequality between agents’ negotiation power) on the correspondence of outcomes achieved by RL agents with the Shapley value. We will also clarify the discussion of alternatives to RL (hand-crafted bots and human daa), and their potential advantages and disadvantages.

---

> > ### Author Response · Authors · 2018-11-22
> > **We revised the paper with a train/test board partition and more boards, an evaluation against a Shapley bot, and a more detailed discussion of the advantages of RL agents versus hand crafted bots**
> >
> > Thanks again for your feedback! As you can see, we have revised the paper:
> >
> > - We now partition boards to a train-set and a test-set, and make sure agents are not memorizing actions for specific boards, and can generalize to previously unobserved boards (i.e. boards not encountered during training). The results regarding the bot-comparison and the Shapley correspondence still hold (see revised figures and numbers in Section 4.2)
> > - We now consider a much larger set of train boards (150 boards) and evaluation boards (50 boards), rather than the 20 boards we had before.
> > - We have added a Shapley-value based bot (similar to the weight-proportional bot, but using a target based on the Shapley value). RL agents are still competitive, even with this more sophisticated bot (which is computationally intractable for games with a much larger number of agents). We also added an experiment regarding training RL agents against a weight-proportional bot, and evaluating them against a Shapley-proportional bot.
> > - We added a discussion regarding the necessity of an RL approach. In short, RL allows us to uncover good negotiation policies, handling diverse negotiation protocols and environments. If one only wants to approximate the negotiation power in an abstract cooperative game, it is sufficient to apply supervised learning (see section 4.4). However such analysis ignores protocol specific details, such as spatial locations, which do affect outcomes (see Figure 4, for example).
> > - We have added experiments regarding the impact of the weight variance (or equivalently, the inequality in negotiation position strength / Shapley values). We find that when the weights have a lower variance (agents are more equal in their negotiation position strength), we get a stronger correspondence with the Shapley values. The new Appendix E contains the full result (with a weight-variance conditional figure equivalent to Figure 4a).

---

### Official Review · AnonReviewer2 · 2018-11-07
**Late Review for Shapley Values Paper**

**Rating:** 5
**Confidence:** 3

**Review:**

Note: This is an emergency review. I managed not to look at existing comments/ratings for this paper before writing my review.

Summary
---

This paper studies deep multi-agent RL in settings where all of the agents must cooperate to accomplish a task (e.g., search and rescue, multi-player video games). It uses simple cooperative weighted voting games 1) to study the efficacy of deep RL in theoretically hard environments and 2) to compare solutions found by deep RL to a fair solution concept known in the literature on cooperative game theory.

In a weighted voting game each agent is given a weight and the agents attempt to form teams. The first team whose total weights exceed a known threshold get the total reward, which is distributed amongst the team members. Given such a game, the __shapely value__ of an agent measures the importance of that agent. How much does it contribute to a team from this set of agents? How much payoff should it get? These have existed in the literature for over 60 years and appear to be widely known and used.

All of this is agnostic to how the agents communicate to form teams: i.e., the communication protocol or the actions available in the environment. The protocol matters because it can allow certain teams to form more or less easily than others, even though the same team would get the same reward regardless of protocol. This can make an agent more or less effective under different protocols. Here two protocols are considered - one where agents suggest proposed teams directly and another where they suggest teams by congregating on a 2d plane. Both protocols result in games whose Nash equilibria are computationally intractable.

The paper shows 4 results:
1) It considers a hand-designed bot similar to models from the game theory literature. Relative to a group of RL agents, an additional RL bot will outputperform a hand-designed bot in terms of average reward it receives.

2) The average reward of a bot is strongly correlated with that bots shapely value.

3) In the negotiation by congregation environment, a bot's spatial position can affect its ability to negotiate.

4) Shapely values can be predicted quite accurately from the weights and threshold that define a cooperative voting game, though these predictions have high variance.

The paper concludes that deep RL is effective at learning agents for cooperative games in multiple ways:
1) Deep agents are better than a hand-designed agent.

2) Deep agents easily extend across negotiation protocols (something hand-designed agents don't do).

3) A popular result in cooperative game theory predicts how effective agents should be. Deep agents are just about that effective.

Strengths
---

* The paper does a pretty good job of reviewing relevant work from game theory.

* Some of the organization is nice (e.g., the list of reasons classic game theory doesn't extend to practice; one section per experiment).

Weaknesses mentioned in individual sections...

Quality
---
Overall, things were well thought through, but I would have liked more out of the experiment 4 section and I think a few minor details might have been missed.

Details:

Section 4.5/Experiment 4: The Shapely value comparison is the most important part of the paper.  This section is important because it tries to explain those results, but it seems like there's more work to be done here. I'm not sure capacity is eliminated as a concern, and there might be other concerns not listed like optimization error.

* I'm not sure what conclusion to take from experiment 4. Shapely values can be computed from the cooperative games directly, independent of protocol. We're interested in __policies__ that get exactly the shapley values as their average reward. Policies depend on the protocol. Does being able to predict shapley values mean that a model with similar capacity can learn a policy that will have the desired shapley value? Was that the desired conclusion?

Other comments:

* The current hand-designed baseline uses weights to form a probability distribution. There should be another baseline that uses Shapley values instead of weights.

* It's not clear exactly what the spatial nature of the Team Patches environment adds. It is good to try another environment just to have an additional notion of generalization.

Clarity
---
Overall, the motivation could be clearer. Is the point to do work on cooperative games or to compare to Shapley values?

Presentation details:

* The paper does not get to specific examples of agents acting in environments until about page 4. Providing a simple, brief example which leaves out some details at the beginning would go a long way toward aiding intuitions about the abstract concepts discussed. Here are some clarity issues I had that might have been helped with an example:
    * What exactly is it about a task which requires agents to form teams? How necessary are those teams?
    * What exactly is a negotiation protocol?
    * What does it mean to distribute/share a reward across agents?

* When talking about shapely values, fairness seems to be emphasized somewhat often, but no concrete intuition about what fairness means in this setting is provided.

* Intro para 4: What does the human data measure? And thus how might it be useful?

* Intro para 7: People in the ICLR community will be more familiar with this work. What is the difference between communication and team forming?

* The section on Shapley values should provide more intuition about what they're thought about as measuring. (An agent's importance or what payoff it should expect, according to wikipedia.)

* Instead of measuring correlation to Shapley values, the paper measures whether average reward approximates Shapley values. It seems like the two are on a different scale. Average reward is unbounded and Shapley values are in [0, 1]. How are they comparable?

* The paper mentions how results vary over different types of boards (ones with higher and lower variance in the sampled weights). It does not show results to support this discussion. A conditional analysis of performance would be interesting and relevant, perhaps conditional versions of Fig. 3.

Originality
---
I do not know much about game theory and I'm only somewhat familiar with multi-agent deep RL, so I am not in a great position to judge novelty. Nonetheless, Given existing work in multi-agent RL, it is unsurprising that deep RL agents learn reasonable policies in these environments.

As far as I know, the comparison of average reward to shapely values has not been done before.


Significance
---
Most work in multi-agent RL evaluates by 1) comparing to baselines or 2) measuring some environment/task-specific metric. The best thing about this work is that it evaluates by comparing actual performance to some external theory that suggests how well an agent should be able to do, falling into a 3rd category.  It's not alone in this category (e.g., paper compare to theoretically optimal baselines if they can), but it is interesting to see another example of this kind of evaluation.

The community might possibly start to focus more on cooperative games because of this paper. A more interesting result would occur if others are inspired to implement more comparisons to how agents __should__ perform in theory.


Justification for Final Rating
---

I am unsure about novelty. As described above, the paper is lacking in clarity and quality (esp. section 4.5), but I don't think these concerns would invalidate the main result. I think the contribution is significant because of the kind of evaluation, but I'm not sure it will ultimately have a large impact. Thus I think some of the concerns above should be addressed before publication, but I would not be very disappointed if it were published as is.

---

> ### Author Response · Authors · 2018-11-12
> **Thanks! We'll add experiments regarding a Shapley bot baseline and weight variance; We'll better discuss the Shapley value and clarify our discussion.**
>
> Thank you for the very thorough and helpful review, especially as an emergency reviewer!
>
> Regarding motivation, our main focus was indeed on comparing the outcomes reached by RL agents with the predictions from cooperative game theory. Cooperative game theory focuses on the negotiation position of players, abstracting away details regarding the specific protocol used to negotiate and share the joint reward. As you point out, when facing a specific protocol, agents seek to maximize their own reward by using an effective policy for that protocol. We show that one can use RL to find effective negotiation policies for any given protocol.
> Regarding novelty, earlier work on multiagent RL has focused on non-cooperative game theory (and in particular on competition between agents or social dilemmas). The key novelty of this work is in comparing how the behaviour of RL agents relates to *cooperative* game theory (which studies how players form teams and share the achieved rewards). To our knowledge we are the first to do so, and we’ll clarify the presentation of our motivation.
>
> As you suggest, we will dedicate more space to discussing the Shapley value as a solution concept. Indeed, the Shapley value range is [0,1], but we measure the *proportion* of the reward an agent achieves on average (which has the same range). The Shapley value is a “power index”, designed to objectively measure the strength of an agent’s negotiating position. It can be viewed as the agent’s power to affect the outcome of the game, or the relative number of opportunities it has to form successful teams. As shown in the example in the appendix, this power is not always proportional to an agent’s weight;  Each agent must infer from experience where their negotiation position lies in the team formation hierarchy, making this an interesting problem in multi-agent reinforcement learning. We’ll make the discussion in the main text longer, and put a more detailed presentation in the appendix.
>
> Regarding a Shapley Bot baseline, we will add a baseline bot using Shapley values rather than proportional weights, and compare with our current agents. This is a much stronger baseline; note that computing Shapley values is NP-hard, so only tractable because we have relatively few agents.
>
> As suggested, we’ll analyze the impact of weight variance, using a conditional version of figure 3 (for high and low variance boards).
>
> Regarding Experiment 4, the experiment shows that given a direct supervision signal (boards labeled with the Shapley values), a small neural net can approximate the Shapley value well. Our RL agents have a more challenging task for two reasons: (1) They have to take into account not only their negotiating position but also protocol details. (2) Their RL supervision is weaker: they only know how successful a whole *sequence* of actions was, and not the “correct” action they should have taken at every timestep. Our conclusion from the experiment is that at least the basic supervised learning task can be accomplished with a small neural network i.e. the agent’s network has the capacity required to estimate their raw negotiating power, abstracting away protocol details. Clearly, there are many further potential reasons for the RL agents to deviate from Shapley (optimization error, incorrect credit assignment and learning dynamics / nonstationarity). Based on your comment we will better motivate the experiment, and briefly discuss the alternative reasons for deviation.
>
> Human data is an alternative to using RL to train agents. Agents can be trained to mimic humans who negotiate under a protocol, but obtaining human data is extremely costly and does not scale.
>
> We proposed the team patches environment to show that our approach generalizes to another negotiation protocol, of a spatial nature. Interacting in the real world requires being at the same physical location at the same time as your negotiation partners. People who negotiate must thus reason about both the high-level negotiation strategy (such as their negotiation position/strength), as well as low-level policies (such as where to go to meet the right partners). We wanted to demonstrate that our approach can handle such complexities. Moreover, just as in the real world, the details of the spatial environment can and do impact the negotiation outcomes in our experiments. We will clarify our discussion of this.
>
> Indeed, we hope this work would convince the community to further investigate RL through a cooperative game theory prism.

---

> > ### Comment · AnonReviewer2 · 2018-11-13
> > **Thanks for the clarifying response.**
> >
> > This addresses most of the main points from my review. It promises a new baseline and improvements to the presentation. Improved presentations of some parts are provided in the response.
> >
> > I still think the paper is missing a larger set of experiments in the vein of experiment 4 to help understand how and why the correlation with Shapley values occurs. The review by AnonReviewer1 shares similar concerns and mentions some potential experiments in the last paragraph.
> >
> > Unfortunately, this isn't quite enough for me to change my rating.

---

> > > ### Author Response · Authors · 2018-11-22
> > > **We have added experiments regarding the correspondence with the Shapley value. We also implemented a train/test partition with more boards, and a bot based on the Shapley value.**
> > >
> > > Thanks again for your feedback!
> > >
> > >
> > > - Indeed, the heart of the paper is the Shapley value comparison. You felt that experiment 4 is incomplete, and that we should add new experiments in the vein of experiment 4 to help understand how and why the correlation with Shapley values occurs. We have added two experiments in that spirit:
> > > 1) an experiment showing how the correlation with the Shapley value depends on the weight variance (or equivalently, the inequality in negotiation position strength); We find that when the weights have a lower variance (agents are more equal in their negotiation position strength), we get a stronger correspondence with the Shapley values: the new Appendix E contains the full result (with a weight-variance conditional figure equivalent to Figure 3).
> > > 2) To further rule out concerns regarding the capacity of RL agents to compute the Shapley value, the new Appendix F has an experiment showing that even when RL agents observe the true Shapley values as a part of the state input, they still deviate from the Shapley value; the experiment indicates that the deviation from Shapley value occurs due to the multi-agent independent RL procedure (agents optimizing for a *policy* that maximizes their *personal gain* in the context of other learners / non-stationary environment).
> > >
> > > - We now partition boards to a train-set and a test-set (with 50 boards instead of 20 evaluation boards we had before), showing that RL  agents can generalize to previously unobserved boards. The results regarding the bot-comparison and the Shapley correspondence still hold (see revised figures and numbers in Section 4.2)
> > > - We have added a Shapley-value based bot (similar to the weight-proportional bot, but using a target based on the Shapley value). RL agents are still competitive, even with this more sophisticated bot. We also added an experiment regarding training RL agents against a weight-proportional bot, and evaluating them against a Shapley-proportional bot.
> > > - We improved the discussion of the motivation and novelty of the work: comparing how the behavior of RL agents relates to *cooperative* game theory (which studies how players form teams and share the achieved rewards).
> > > - We expanded the discussion of the Shapley value, and why it measures the strength of an agent’s negotiation position, or the fair share of the reward it should receive. We now provide a list of the fairness axioms and related work on power indices (see Appendix A)
> > >
> > >
> > > We still have a couple of days to further revise the paper, so any suggestions you have after reading the revised paper are very much appreciated!

---

### Meta-Review · Area_Chair1 · 2018-12-17

**Confidence:** 4
**Recommendation:** Reject

**Metareview:**

This paper was reviewed by three experts. Initially, the reviews were mixed with several concerns raised. After the author response, there continue to be concerns about need for significantly more experiments. If this were a journal, it is clear that recommendation would be "major revision". Since that option is not available and the paper clearly needs another round of reviews, we must unfortunately reject. We encourage the authors to incorporate reviewer feedback and submit a stronger manuscript at a future venue.